# A 24 GHz End-Fire Rod Antenna Based on a Substrate Integrated Waveguide

**DOI:** 10.3390/s25051636

**Published:** 2025-03-06

**Authors:** Yanfei Mao, Shiju E, Yu Zhang, Wen-cheng Lai

**Affiliations:** 1Engineering College, Zhejiang Normal University, Jinhua 321000, China; yfmao@zjnu.cn (Y.M.); eshiju@163.com (S.E.); 2Laboratory of Urban Rail Transit Intelligent Operation and Maintenance Technology & Equipment of Zhejiang Province, Zhejiang Normal University, Jinhua 321000, China; 3Department of Electrical Engineering, Ming Chi University of Technology, New Taipei City 243303, Taiwan

**Keywords:** antenna, end-fire, planar, rod, SIW

## Abstract

Most of the traditional rod antennas in the literature are in the shape of a cylinder or are conical, which are not suitable shapes for planar PCB technology or planar integrated CMOS or BiCMOS technology. In this paper, we present a 24 GHz planar end-fire rod antenna based on an SIW (substrate integrated waveguide) suitable for planar PCB technology or planar integrated circuit technology. The antenna is made of PCB Rogers 4350 and utilizes the SIW to realize the end-fire rod antenna. The measurement results of the antenna are presented: its gain is 8.55 dB and its S11 bandwidth is 6.2 GHz. This kind of planar end-fire rod antenna possesses the characteristics of high gain, wide bandwidth, compactness, and simple design and structure. This type of antenna can also be used as a PCB antenna in other frequency bands, and it could also possibly be utilized in mm-wave and THz integrated antenna design in the future due to its very simple architecture.

## 1. Introduction

According to reference [1], comparing an end-fire antenna array with a broadside array, the parameters of an end-fire antenna array, like the half-power beam width and directivity, are all twice that of broadside antenna arrays. Therefore, the authors have carried out research on an end-fire antenna over the past several years [2,3,4]. In this paper, the authors present and discuss a 24 GHz end-fire rod antenna based on an SIW, which is made of PCB Rogers 4350.

In RF circuit design, there is a default design rule: the simpler, the better. Based on this rule, the authors tried to find a planar end-fire antenna architecture based on an SIW that can be utilized in PCB antenna design as well as integrated millimeter-wave or THz integrated antenna design in the future [5]. With regard to integrated antennas, with reference to the 245 GHz integrated on-chip antenna in [5], we can see that for integrated antennas, a simple architecture is very important because a complicated architecture will surely require many more steps in IC fabrication procedures and increase the cost of IC technology dramatically. Therefore, an end-fire antenna with a very simple architecture is a design challenge in integrated THz antenna design.

Most of the traditional end-fire rod antennas in the literature [6,7,8,9,10,11,12,13,14,15,16,17,18] are in the shape of a cylinder or are conical. Reference [6] demonstrated the UWB and stable/symmetric pattern properties of a three-layer rod antenna in the shape of a cylinder at 18 GHz. Reference [7] presents a beam-adjustable multibeam dielectric rod antenna for mm-wave RF-WPT applications; three or five rods are utilized for each antenna, and a gain value above 12 dBi is maintained between 20 GHz and 24 GHz. All three or five rods are in the shape of a cylinder. Reference [8] discusses a polyrod antenna, which is fed by a tapered input region extended into a circular waveguide. Reference [9] discusses an end-fire rod antenna with coaxial feed, impedance tuning, and a reflected ground at input. The rod antenna is fed by a circular waveguide, and the material of the rod is polystyrene. Reference [10] presents a wideband dielectric rod (polyrod) antenna operating from 7 to 18 GHz, and the rod is in the shape of a cylinder. In reference [11], the effect of the surface roughness of dielectric rod antennas on radiation performance is analyzed, and the dielectric rod antenna consists of a cylindrical structure and a frustum structure. Reference [12] presents an ultra-wideband dielectric circular rod antenna using a novel V-shaped twin-wire tapered transverse electromagnetic waveguide as the feed structure, with a gain of 17 dB. In reference [13], a patch antenna with a dielectric circular rod is proposed, whose gain is 13.5 dB at 60 GHz. Reference [14] presents dielectric rod antennas in a focal plane array for terahertz imaging at 600 GHz, and the rods are in the shape of a cylinder. In reference [15], a new twin-coil ferrite circular rod (TCFR) antenna for harvesting RF energy in the AM band is proposed. In reference [16], a dielectric rod antenna with small aperture, high gain (higher than 10 dB), and a wide frequency band is realized by introducing parallel coaxial stepped block feed and an optimized dielectric rod. In reference [17], for the existing terrestrial broadcast services AM, FM, and DAB (VHF and L-Band), an active rod antenna with a new low-cost design and a height of only 14 cm is presented. A novel antenna integrated the advantages of the Yagi–Uda antenna, and the dielectric rod antenna (DRA) is presented in reference [18]. Reference [19] proposes a reconfigurable lower RCS antenna that is implemented using a ring-shaped ferrite rod array and a line source, and a cylindrical rod antenna was utilized in the rod array. The effect of displacement errors in dielectric rod antenna arrays on radiation performance was analyzed in [20], and a rod antenna in the shape of a cylinder was adopted in [20]. Reference [21] presents the design and experimental validation of a novel dielectric rod electromagnetic band-gap (EBG) leaky-wave antenna (LWA) that operates in the K-band. In [22], tunable radiation from a periodic chain of graphene-coated dielectric rods is investigated by using a rigorous ad hoc full wave modal solver. Reference [23] proposed a new high-gain dielectric rod antenna (DRA) operating in the Ka-band, with a gain of 21 dBi.

References [24,25,26,27,28,29,30,31,32,33,34] present some examples of planar end-fire rod antennas. Reference [24] presents the simulation results of an S-band dielectric rod antenna based on substrate integrated waveguide (SIW) technology, with a gain of 8.5 dB at 3.5 GHz. Reference [25] presents a rectangular dielectric rod antenna (RDRA) 3D printed using the liquid crystal display resin method, with a gain of 12.7 dB at 24 GHz. In reference [26], a new broadband high-gain planar dielectric rod SIW antenna is proposed, and the antenna has a simulated gain of 9–13 dB from 24 to 50 GHz. Reference [27] presents a 135 GHz Yagi dipole rod antenna-in-package with a gain of 10.3 dB. Reference [28] presents a wideband high-gain dielectric lens integrated with a tapered rod antenna using a perforated H-guide for W-band [(75 to 110) GHz] applications, and an extended hemisphere dielectric lens is integrated with the radiating tapered end. Reference [29] presents a structurally integrated design of a dielectric flange and a dielectric rod antenna (DRA) in the millimeter-wave band, with a gain of 10.5 dB (sim) at 28 GHz. Reference [30] presents a broadband 3D-printed dielectric rod antenna, which is fed by an N-type coaxial connector that excites a ridge waveguide as the feed source, with a gain of 14.9 at 11 GHz. In reference [31], a dielectric rod antenna (DRA) with inexpensive 3D printing processes is proposed for millimeter-wave (mm-wave) applications, and the antenna has a gain of 20 dB at 30 GHz. Reference [32] presents a dielectric rod antenna in the x-band, and the proposed structure exhibits a bandwidth of 4% at 10 GHz and a peak gain of 9 dBi in the operating band for a 1λ_0_ long dielectric rod radiator. Reference [33] reports interesting dielectric rod waveguide-coupled CMOS-based sources and detectors, and a dielectric rod-coupled system enables it to reach a >60 dB signal-to-noise ratio for an equivalent noise bandwidth of one Hz. In reference [34], a microstrip-line-excited ultra-wideband dielectric rod antenna manufactured using 3D printing technology is presented, with a gain of 13.3 dB at 11 GHz.

This paper presents a planar end-fire rod antenna made of PCB Rogers 4350. This end-fire planar rod antenna has a very simple architecture and meets the requirements of the challenges of on-chip antenna design. Therefore, it is possibly suitable for applications like integrated THz antenna design, and it is of course also suitable for PCB antenna design in other frequency bands. A patent is under application as well for this architecture of a planar end-fire antenna based on an SIW.

## 2. Topology of the Rod Antenna

Figure 1a is the 3D view of the rod antenna. We can see that the 24 GHz end-fire rod antenna is composed of four parts: (1) an input microstrip, (2) a substrate integrated waveguide (SIW), (3) two tapered triangles—they are the transition between the input microstrip and the SIW—and (4) the bare dielectric rod. The three-dimensional view of the antenna includes the GSG coplanar waveguide (the GSG structure). In Figure 1a, all of the metal layers are in red, and all of the vias are in yellow. The remaining part is the dielectric Rogers 4350.

The 3D view of the antenna without the GSG structure is also shown in Figure 1b. In order to clearly describe the topology of the antenna, the top view and bottom view of the antenna are also shown in Figure 2a,b. In Figure 1b and Figure 2a,b, the shaded part is metal copper, while the other remaining part is Rogers 4350.

The dielectric rod is a trapezoid shape. The transition between the input coplanar microstrip and the SIW is two tapered triangles in order to realize impedance matching between the microstrip and the SIW.

The main parameters of the prototype of the end-fire rod antenna are shown in Table 1. The thickness of the antenna dielectric: Rogers 4350 H1 is 3.8 mm. In order to obtain a dielectric thickness of around 3.8 mm, three layers of Rogers 4350 with thicknesses of 1.524 mm, 1.524 mm, and 0.76 mm, respectively, are adhered to one another.

The working principle of the antenna is as follows: The RF energy flow in the microstrip is a TEM wave. The main RF energy flow in the SIW is TE10 mode. Electric fields are continuous at the interface of the microstrip and SIW; therefore, two tapered triangles realize impedance matching between the two. The length of SIW L3 is a half wavelength at 24 GHz. The SIW constrains RF energy, transfers energy, and forms the end-fire emission of the RF energy through the bare dielectric rod, as shown in Figure 3. Figure 3 shows the E field animation plot of the end-fire rod antenna in HFSS. As shown in Figure 3, there is one repeated parcel inside the SIW structure, and it shows that the length of SIW L3 is a half wavelength. If the length of the SIW is one wavelength, then it will contain two repeated parcels in the E field animation. The SIW constrains the RF energy flow, transfers the energy flow, and forms the end-fire emission of the RF energy through the bare dielectric rod, whether the rod is one kind of tapered shape, as shown in Figure 4, a triangle, a trapezoid, or a curved one. In this paper, the rod is a trapezoid.

The permittivity of the dielectric Rogers 4350 is greater than air. At the interface of Rogers 4350 and the air, RF signals obey Snell’s laws of reflection and refraction; therefore, RF signals tend to focus rather than diverge when it is radiating through the tapered dielectric rod, and this is also another theoretical foundation for this end-fire rod antenna. Figure 5 shows the reflection and refraction of parallel light passing through the trapezoid (the model in Figure 5 is not accurate when the dimensions are close to the wavelength, but it can provide a more intuitive understanding). The blue arrows represent the parallel RF energy flow, the yellow arrows represent the reflected waves, and the green arrows represent the refracted waves. According to Snell’s laws of reflection and refraction, regardless of whether they are reflected or refracted, such waves tend to concentrate and focus compared with the input parallel waves. For triangular and curved rods, the principles are similar.

The main parameters in Table 1 are obtained through optimizations in HFSS time and again as follows:(1).According to reference [35], for the SIW-only single TE10 mode to propagate, 2W3>λ>{W3,2H1}, usually $H1=\left(0.4\sim 0.5\right)W3$. And in this case, H1/W3 is around 0.35. Thickness H1 should adapt to the feasibility of the thickness of Rogers 4350 on the market. Therefore, values of H1 are limited combinations of thickness available on the market, like “0.101 mm, 0.254 mm, 0.508 mm, 0.762 mm, 1.524 mm” and so on. H1 can be optimized easily due to its limited choices. Finally, we chose H1 = 1.524 + 1.524 + 0.76 mm due to its excellent performance in TE10 mode at 24 GHz.(2).After H1 is determined, W3 can also be determined due to the relationship of H1 = (0.4~0.5) × W3 mentioned in (1). When W3 increases, the SIW will reach the multi-resonance region; when W3 decreases, SIW will have the best performance for the single TE10 mode at 24 GHz; and when W3 decreases further, the resonance frequency will shift up, and become greater than 24 GHz. In this way, W3 is optimized to be 11 mm for the single TE10 mode at 24 GHz. When W3 is equal to 11 mm, the distance between vias holes is 9.15 mm. Inside Rogers 4350 (D_k_ = 3.48), the wavelength at 24 GHz is 6.7 mm. Therefore, TE30 will not exist. Moreover, the symmetry of the antenna will be less likely to excite mode TE20. Therefore, the SIW still works in wide TE10 mode.(3).The length of SIW L3 is around a half wavelength. It is optimized and determined when the SIW contains one repeated parcel in the E field animation plot of HFSS.(4).The gain of this rod antenna can be modified by changing the length of the dielectric rod L4. The gain increases when L4 increases, but it saturates when L4 increases further, greater than around 17 mm. Therefore, in this paper, L4 was set as 17.2 mm. Width W4 was optimized simultaneously with L4 to obtain the best gain performance.(5).The size of the trapezoid W2, L2 is optimized to obtain the best S11 performance of the entire antenna.(6).The length of the input microstrip L1 was determined according to the size of the 2.92 mm end launch connector fabricated by Qualwave Inc., Chengdu, China. L1 should be large enough to install the 2.92 mm end launch connector.(7).As a matter of fact, the input GSG structure affects antenna performance a lot. The input GSG structure is optimized by fabricating and measuring the antenna twice. Optimization of the input GSG structure is as follows.

During the first fabrication and measurement of the rod end-fire antenna, an input GSG structure with small vias similar to the antennas in [2,3,4] is utilized, as shown in Figure 6.

Figure 6 shows the top view, bottom view, and side view of the prototype of the end-fire rod antenna during the first fabrication and measurement. As shown in Figure 6a, a GSG coplanar waveguide with four small rectangular vias is designed as input for the installation of the end launch connector during measurement. To measure the end-fire rod antenna, a 2.92 mm end launch connector fabricated by Qualwave Inc., China, was used, as shown in Figure 6a.

The input GSG structure with four rectangular vias limits the bandwidth a lot, as shown in Figure 7. The end-fire rod antenna achieves a 3 dB bandwidth of S11, with merely 1.9 GHz. It is a very narrow bandwidth. The measured gain of the rod end-fire antenna during the first fabrication is 8.27 dB. Therefore, in order to obtain a rod end-fire antenna with a wider bandwidth, the authors redesigned the input GSG structure, and Figure 8 shows the new input GSG structure for the second fabrication of the rod antenna.

Figure 8 shows the top view, bottom view, and side view of the prototype of the end-fire rod antenna during the second fabrication and measurement. As shown in Figure 8a, a GSG coplanar waveguide without the four small rectangular vias was designed as input for the installation of the 2.92 mm end launch connector during measurement. With the new input GSG structure during the second fabrication, this type of end-fire rod antenna achieves a much wider bandwidth, 6.3 GHz (from 22.2 GHz to 28.5 GHz, S11 < −10 dB), as shown in Figure 9.

## 3. Simulation and Measurement Results of the Rod End-Fire Antenna

The measurement and simulation results of S11 and the gain pattern of the antenna during the second fabrication and measurement are presented in Figure 9, Figure 10 and Figure 11.

The input GSG structure affects S11 performance a lot. Therefore, in Figure 9, the S11 simulation results with/without the new GSG structure are both shown. And the measured S11 with the new GSG structure is also shown. Compared with the S11 simulation results with/without the input GSG structure, the resonance frequency shifted from 24 GHz to around 27.5 GHz. This might be due to the inclusion of the end-launch connector in the measurement.

In Figure 10 and Figure 11, with the new input GSG structure in the second fabrication, this type of end-fire rod antenna achieves a much wider bandwidth: 6.3 GHz (from 22.2 GHz to 28.5 GHz, S11 < −10 dB). It obtains a maximum gain of 8.55 dB. In the x-y plane, the half-power bandwidth is 41°, and in the x-z plane, the half-power bandwidth is 47°. In Figure 10 and Figure 11, the simulated gain is 10.9 dB. We can see that the gain drops from 10.9 dB in the simulation to 8.55 dB in the measurement. The great drop in gain might be due to the GSG CPW grounds being too close to the SIW launching triangle, thus creating short tapered slots pointing in +Y and -Y directions. The parasitic radiation from the slots is probably the cause of the reduction in the measured gain vs. the simulated gain. In the next version, the GSG structure should be further away from the SIW launching triangle. And for possible application in integrated antenna design, the GSG structure will not be needed at all. And all of the errors introduced by the GSG structure, like the shift of the resonance point, gain reduction, and so on, will disappear. The drop in gain might also be due to fabrication tolerances or connector losses and so on.

The cross-polarization pattern is also included in Figure 10 and Figure 11. The measured cross-polarization is smaller than −30 dB in both x-y and x-z planes.

As shown in Figure 9, Figure 10 and Figure 11, the measurement results of this rod antenna mainly agree with the simulation results.

The radiation efficiency is 0.58, and the sidelobe level in the x-y plane is −5 dB.

As shown in Figure 11, the measured maximum gain in the x-z plane tilts with a degree of 7° toward the -z direction compared with the simulation results. Tilting of the beam might be due to the inclusion of the GSG structure at the input and the error in the angle of positioning of the antenna in the measurement. In practical applications, a 7° beam tilt might reduce the communication distance between the receiver and the transmitter.

## 4. Discussion

Table 2 compares the end-fire rod antenna with other, earlier rod antennas [24,25,26,27,28,29,30,31,32,33,34]. The antennas in [24,25,26,27,28,29,30,31,32,33,34] are all end-fire rod antennas at different frequencies. The rod antenna in [24] utilizes three kinds of dielectrics to realize the function of the antenna based on the SIW; the structure is complicated, while the rod antenna in this work is much simpler in design. The rod antenna in [25] is a bare dielectric rod, utilizing external waveguide transition to realize the function of the waveguide; it utilizes Al_2_O_3_ to modify the permittivity of the dielectric material, and the integration level of the rod antenna in [25] is low. It could not be designed as an on-chip antenna in the future. Comparing the size of the two antennas, at the same working frequency of 24 GHz, the antenna in this work is much more compact than the antenna in [25]. The antenna in [26] utilizes an additional antipodal Vivaldi to radiate energy fed in by the SIW, rather than the rod antenna alone, and achieved a gain of 11 dB at 36 GHz. In order to design the rod antenna, two different kinds of dielectric materials with different permittivity are utilized in [26]. The rod antenna in [27] is an antenna-in-package for a 135 GHz transmitter; the antenna utilizes an additional dipole unit to radiate energy through the rod antenna as well. Compared with the rod antennas in [26,27], the antenna in this work has a lower gain but a simpler design and structure. The antenna in [28] has a much higher gain because it includes an additional huge dielectric lens to enhance gain, and it also utilizes heavy external metal housing to realize the function of the perforated H-guide. Compared with the antenna in [28], the antenna in this work has a lower gain but is much simpler in structure; also, the antenna in [28] is not suitable for on-chip planar antenna design due to its cylinder dielectric lens. In [29], a structurally integrated design of the dielectric flange and dielectric rod antenna (DRA) is implemented in the millimeter wave band with a gain of 10.5 dB at 28 GHz. Comparing the sizes of the two antennas in [29] and in this work, at the similar working frequencies of 24 and 28 GHz, the antenna in this work is much more compact than the antenna in [29], and the antenna in [29] is not a candidate as a planar on-chip antenna design due to its use of a flange. In [30,34], the 3D printing process is utilized for a high gain wideband dielectric rod antenna for mm-wave applications. Nevertheless, after careful checks, the antennas in [30,34] have curve lines in vertical directions in the antenna design; therefore, they are, in principle, not planar rod antennas and might not be candidates as integrated antenna designs in planar BiCMOS or CMOS technology. A dielectric rod antenna (DRA) with an inexpensive 3D printing processes is proposed in [31], working at similar frequencies of 24 and 30 GHz; the gain of the antenna in [31] is higher than that of the antenna in this work, but the antenna in this work is much more compact than the rod antenna in [31].

To sum up, this planar end-fire rod antenna based on the SIW is characterized by appropriate gain, a wide bandwidth, and a simple design and structure and shows some research value. In particular, this type of planar end-fire antenna architecture based on the SIW meets the requirement that on-chip antennas demand a simple structure and it is possibly suitable for applications like integrated planar THz antenna design.

## 5. Conclusions

A 24 GHz end-fire planar rod antenna based on an SIW is presented and discussed; the measured gain is 8.55 dB. This type of end-fire rod antenna is made of PCB Rogers 4350. It possesses characteristics like high gain, wide bandwidth, simple design, compactness, and so on. The topology of this end-fire rod antenna can of course be utilized in PCB antenna design in other frequency bands. Moreover, this type of planar end-fire antenna architecture based on the SIW meets the requirement that on-chip antennas demand simplicity in structure, and it is possibly suitable for applications like integrated planar THz antenna design, when the frequency increases and its size decreases.

## Figures and Tables

**Figure 1 sensors-25-01636-f001:**
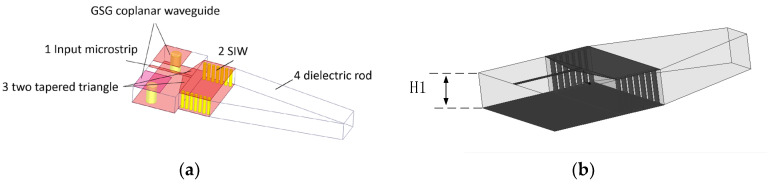
(**a**) Architecture of the end-fire rod antenna and (**b**) 3D view of the rod antenna without the GSG structure.

**Figure 2 sensors-25-01636-f002:**
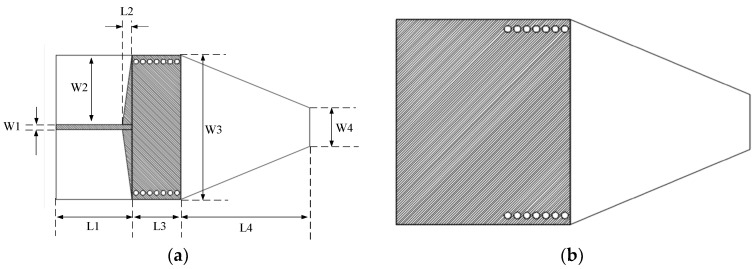
(**a**) Top view of the rod antenna and (**b**) top view of the antenna.

**Figure 3 sensors-25-01636-f003:**
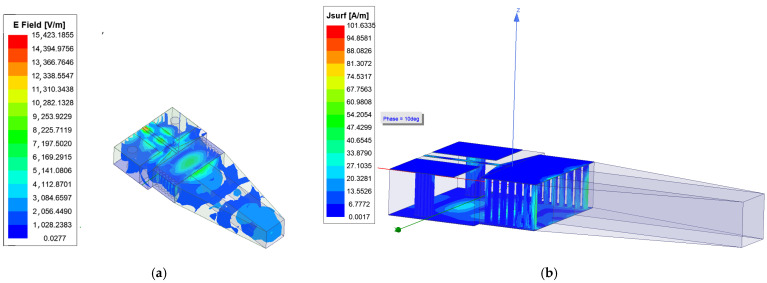
(**a**) E field animation plot and (**b**) current plot (Jsurf plot) of the end-fire rod antenna in HFSS.

**Figure 4 sensors-25-01636-f004:**
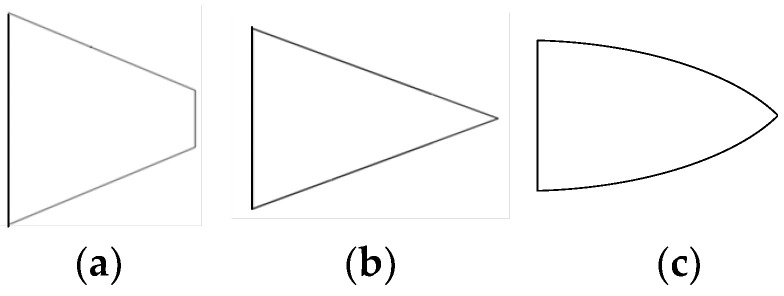
The rod can be one kind of tapered shape: (**a**) trapezoid, (**b**) triangle, or (**c**) curved.

**Figure 5 sensors-25-01636-f005:**
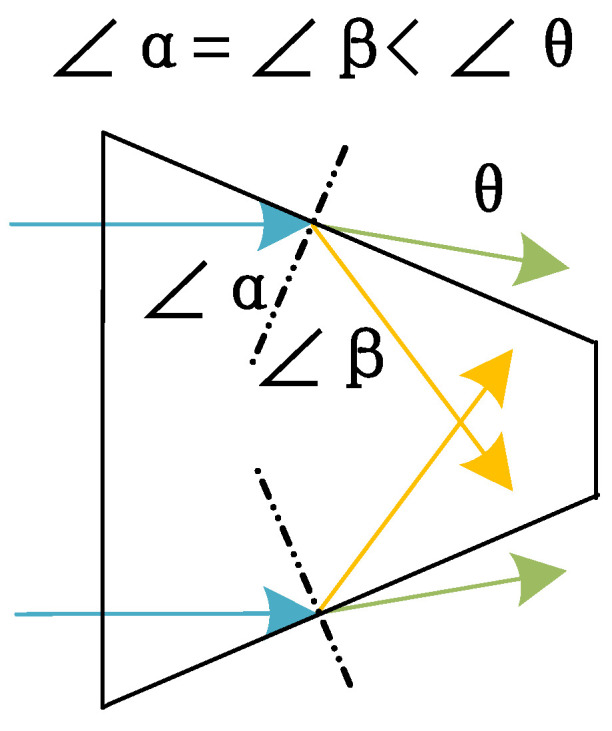
Explanation of the reflection and refraction at the interface of the rod and air according to Snell’s law.

**Figure 6 sensors-25-01636-f006:**
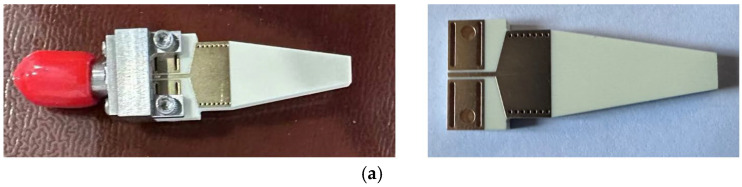
(**a**) Top view, (**b**) bottom view, and (**c**) side view of the prototype of the end-fire rod antenna during the first fabrication (the coordinate *X*-*Y*-*Z* axis is also shown in (**c**)).

**Figure 7 sensors-25-01636-f007:**
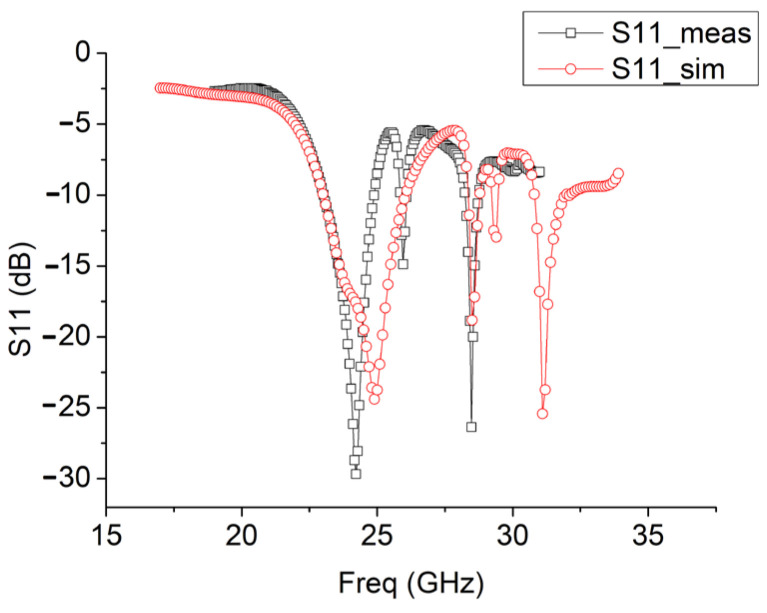
Simulation and measurement results during the first fabrication of the rod end-fire antenna.

**Figure 8 sensors-25-01636-f008:**
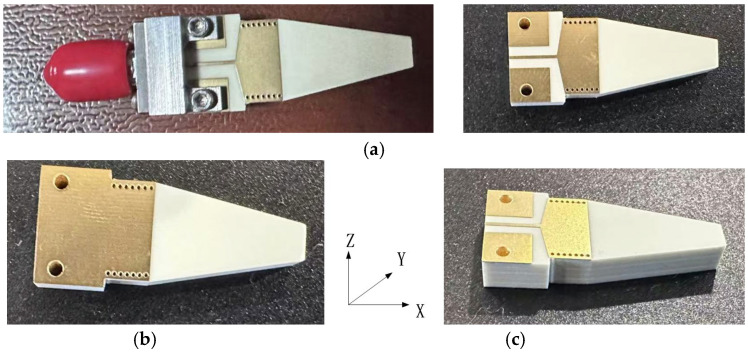
(**a**) Top view, (**b**), bottom view, and (**c**) side view of the prototype of the end-fire rod antenna during the second fabrication (the coordinate *X*-*Y*-*Z* axis is also shown in (**c**)).

**Figure 9 sensors-25-01636-f009:**
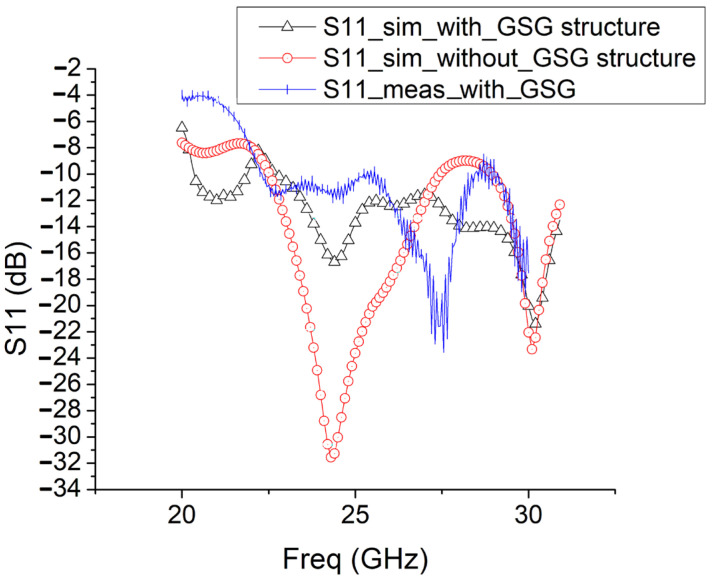
Simulation and measurement results of the S parameter of the antenna during the second fabrication and measurement.

**Figure 10 sensors-25-01636-f010:**
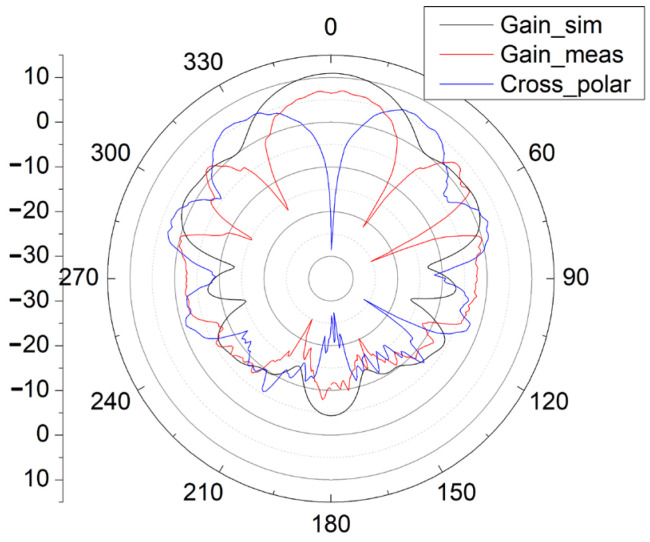
Simulation and measurement results of the gain pattern of the antenna in the x-y plane (gain versus φ).

**Figure 11 sensors-25-01636-f011:**
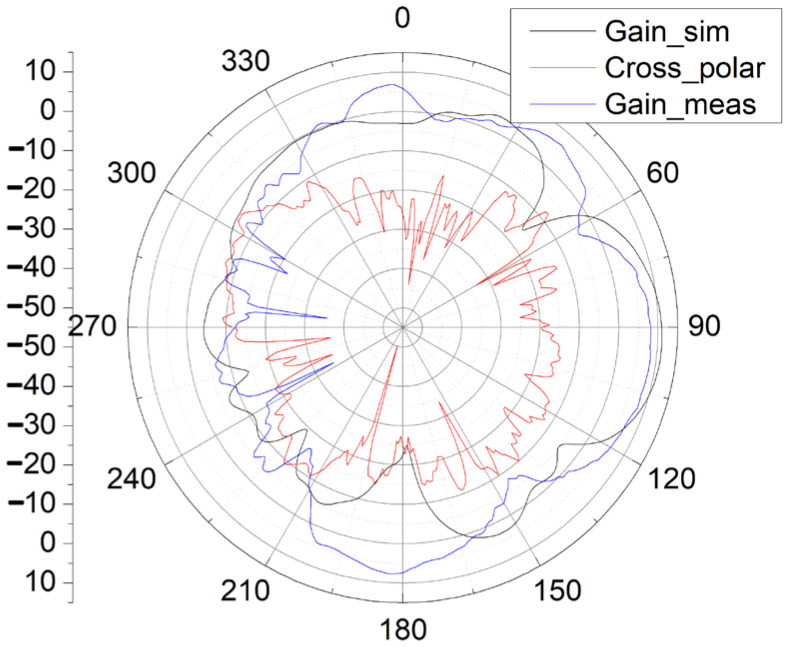
Simulation and measurement results of the gain pattern of the antenna in the x-z plane (gain versus θ).

**Table 1 sensors-25-01636-t001:** Parameters of the prototype of the end-fire rod antenna.

Parameters	Values (mm)
W1	0.5
W2	5.25
W3	11
W4	4
H1	3.8
L1	7.8
L2	1
L3	5
L4	17.2

**Table 2 sensors-25-01636-t002:** Comparison table of the present work with earlier designed structures.

	Working Freq (GHz)	Antenna Gain (dBi)	Size	Bandwidth (GHz)	Architecture
[24]	3.5	8.5 (sim)	24.8 cm × 62.4 × 1.36 cm	3.14–3.8 (sim)	Dielectric rod antenna based on an SIW
[25]	24	12.7	69.6 mm × 10.7 mm × 6 mm	19.5–28.5	3D-printed dielectric rod antenna for surface wave manipulation
[26]	36	11 (sim)	44.75 mm × 9.5 mm × 0.058 mm	24–50 (sim)	Dielectric rod antenna with an antipodal Vivaldi based on an SIW
[27]	135	10.3	N/A	128–142	Rod antenna with a Yagi dipole unit-in-package
[28]	92.5	23.9	65 mm × 29 mm	75–100	Dielectric lens integrated with a tapered rod antenna using a perforated H-guide
[29]	28	10.5 (sim)	50 mm × 50 mm × 100 mm	26.5–30.5	A structurally integrated design of the dielectric flange and dielectric rod antenna (DRA)
[30]	11	14.9	20.3 mm × 39 mm × 163 mm	6–16	A broadband 3D-printed dielectric rod antenna
[31]	30	20 (sim)	>150 mm	26–40	A dielectric rod antenna (DRA) with inexpensive 3D printing processes
[32]	10	9	40.2 mm × 35.5 mm × 30 mm	9.71–10.21	A substrate integrated waveguide (SIW)-based band pass filter is used to feed the dielectric rod through a slot
[34]	11	13.3	130 mm × 40 mm × 15 mm	6–16	A microstrip-line-excited ultra-wideband dielectric rod antenna manufactured using 3D printing technology
This work	24	8.55	30 mm × 13 mm × 3.8 mm	22.2–28.5	A rod antenna based on a SIW

## Data Availability

The raw data supporting the conclusions of this article will be made available by the authors upon request.

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
