# Peer review of "A 24 GHz End-Fire Rod Antenna Based on a Substrate Integrated Waveguide"

_sensors, 2025, doi:10.3390/s25051636_

Round 1
Reviewer 1 Report
Comments and Suggestions for Authors
The article presented a well-designed 24 GHz end-fire rod antenna based on SIW, with clear advantages in terms of simplicity, compactness, and suitability for planar integration. The methodology is almost clear, and the results are well-supported by simulations and measurements. However, the paper could be improved by providing more critical analysis of the design choices, a deeper discussion of the results, and a clearer emphasis on the novelty of the proposed antenna. Herein are some comments can improve the quality of the results:
*How were the dimensions of the trapezoidal rod (W4, L4) optimized? Were there any trade-offs between gain, bandwidth, and size during the optimization process? *The discussion should be more deeply and sharpe between the simulated and measured gain (10.9 dB vs. 8.55 dB). *Could consider other potential factors, such as fabrication tolerances or connector losses. Additionally,could discuss the impact of the 7° beam tilt observed in the measurement and how it might affect practical applications.*The comparison table (Table 2) is good, but should be more specific and discussion could be more critical in the Architecture part. In addition,should highlight the design's suitability for specific applications, such as integrated THz antennas.
OK
Author Response
The article presented a well-designed 24 GHz end-fire rod antenna based on SIW, with clear advantages in terms of simplicity, compactness, and suitability for planar integration. The methodology is almost clear, and the results are well-supported by simulations and measurements. However, the paper could be improved by providing more critical analysis of the design choices, a deeper discussion of the results, and a clearer emphasis on the novelty of the proposed antenna. Herein are some comments can improve the quality of the results:
Comments 1,
*How were the dimensions of the trapezoidal rod (W4, L4) optimized? Were there any trade-offs between gain, bandwidth, and size during the optimization process?
Response : Dear reviewer, thank you sincerely for your suggestions. W4 and L4 are optimized as follows: Gain of this rod antenna could be modified by changing the length of the dielectric rod L4. Gain increases when L4 is increasing, but it will saturates when L4 increases further, greater than around 17 mm. Therefore, in this paper L4 is set as 17.2 mm. Width W4 is optimized simultaneously with L4 to obtain best gain performance.
Gain is mainly determined by length of L4. The longer is the rod L4, the gain is bigger. But the gain will saturate when L4 is too long. Bandwidth is optimized mainly by the input matching, e.g. the input GSG structure , the two tapered triangle, and length L1, width of the microstrip W1 and so on. The wider is the antenna width W3, the wider is the bandwidth. And when W3 is too big, it will reach multi-resonance region. In a word, when size L4 increases, gain increases, and gain will saturate when L4 is too long. When width W3 increases, bandwidth increases, and bandwidth will reach multi-resonance region if W3 is too big.
Comments2,
*The discussion should be more deeply and sharpe between the simulated and measured gain (10.9 dB vs. 8.55 dB).
Response: Dear reviewer, thank you sincerely for your suggestions.
I have included the factors” fabrication tolerances or connector losses” in the paper, in line 232, as follows, marked with red color:
“The drop in gain might be also due to fabrication tolerances or connector losses and so on.”
Comments 3,
*Could consider other potential factors, such as fabrication tolerances or connector losses. Additionally, could discuss the impact of the 7° beam tilt observed in the measurement and how it might affect practical applications.
Response: Dear reviewer, thank you sincerely for your suggestions. I have included the factors” fabrication tolerances or connector losses” in the paper, in line 232, as follows, marked with red color:
“The drop in gain might be also due to fabrication tolerances or connector losses and so on.”
In line 242, discussion about 7° beam tilt is given as follows, marked with red color:
“In practical applications, 7° beam tilt might reduce the communication distance of between the receiver and transmitter. “
Comments 4,
*The comparison table (Table 2) is good, but should be more specific and discussion could be more critical in the Architecture part. In addition, should highlight the design's suitability for specific applications, such as integrated THz antennas.
Response: Dear reviewer, thank you sincerely for your suggestions.
In line 281, I add some words to highlight the design's suitability for specific applications, such as integrated THz antennas, as follows, marked with red color:
“Especially this type of planar end-fire antenna architecture based on SIW meets the requirement that on-chip antennas demand simplicity in structure and it is possibly suitable for applications like integrated planar THz antenna design.”

Reviewer 2 Report
Comments and Suggestions for Authors
Please consider revising the paper and focus more on improving the English and readiability of the manuscript.
1. Please revise first page of introduction. Also, second page of introduction needs major English revisions.
2. Authors are requested to provide more comprehensive view of the design, equations, physics, equivalent circuit model or plots of current distributions.
3. In line 127, “.. then it will contains….” It should be contain.
4. Use equation editor in line 152.
5. Sketch of Fig 5 is not clear. Please provide better and clear sketches.
6. Some discrepancies in Fig 7 is observed above 30 GHz. Can author comment on this in the results?
7. There are major issues in Fig 9, where discrepancy between simulated and measured results are seen. Authors need to provide justifications for this.
Comments on the Quality of English LanguageThe quality of English language in the manuscript is really poor. Authors need to improve on the quality of the paper in terms of English, using proper grammar...etc.
Author Response
Please consider revising the paper and focus more on improving the English and readability of the manuscript.
- Please revise first page of introduction. Also, second page of introduction needs major English revisions.
Response: Thank you sincerely for your suggestions.
- Authors are requested to provide more comprehensive view of the design, equations, physics, equivalent circuit model or plots of current distributions.
Response: Thank you sincerely for your suggestions.
In figure 3, I also include the Jsurf plot of the antenna, which is marked with red color.
- In line 127, “.. then it will contains….” It should be contain.
Response: Thank you sincerely for your suggestions. Now I have already changed “contains” into “contain”.
- Use equation editor in line 152.
Response: Thank you sincerely for your suggestions. Now I have already used the equation editor to write the equations in line 152, which is marked with red color.
- Sketch of Fig 5 is not clear. Please provide better and clear sketches.
Response: Thank you sincerely for your suggestions. Now I have make the words and lines in picture 5 thicker and the figure is also bigger. Sincerely hope that now it is better.
- Some discrepancies in Fig 7 is observed above 30 GHz. Can author comment on this in the results?
Response: Thank you sincerely for your suggestions. Figure 7 shows simulation and measurement results in 1st fabrication of the rod end fire antenna. The GSG structure of the antenna in figure 7 is shown as below. The GSG structure is not adopted in the final version of the antenna because it reduced the bandwidth a lot. From figure below we can see the GSG structure is very complicated with 4 square via holes which connect the top ground metal with the bottom ground metal. I think discrepancies are introduced by the complicated GSG structure.
- There are major issues in Fig 9, where discrepancy between simulated and measured results are seen. Authors need to provide justifications for this.
Response: Thank you sincerely for your suggestions. In figure 9, compared with S11 simulation results with/without input GSG structure, the resonance frequency shifted from 24 GHz to around 27.5 GHz. This might be due to inclusion of the end-launch connector in measurement, and also it might be due to fabrication tolerances or connector losses and so on.

Reviewer 3 Report
Comments and Suggestions for Authors
The paper presents a 24 GHz end-fire rod antenna based on SIW technology. The topic is interesting. Some suggestions are given below.
- Please provide more details on the limitations of traditional rod antennas and the specific challenges addressed by the proposed design in the introduction.
- The authors should clarify additional performance metrics, such as radiation efficiency and sidelobe levels, in the results section.
- More recent references should be added.
- Figures are not clear enough.
- The writing style is not good.
- What is the meaning of Figure 8? Please clarify it.
Author Response
The paper presents a 24 GHz end-fire rod antenna based on SIW technology. The topic is interesting. Some suggestions are given below.
- Please provide more details on the limitations of traditional rod antennas and the specific challenges addressed by the proposed design in the introduction.
Response: Thank you sincerely for your suggestions.
Most of the traditional rod antennas in references are in the shape of cylinder or conical which are not suitable for planar PCB technology or planar integrated CMOS or BiCMOS technology, and this paper presents a 24 GHz planar end-fire rod antenna based on SIW (substrate integrated waveguide) suitable for planar PCB technology or planar integrated circuit technology.
For integrated antenna, simple architecture is very important, because complicated architecture will surely require many more steps in procedures of IC fabrication and improve the cost of IC technology dramatically. Therefore an-end fire antenna with very simple architecture is a design challenge for integrated THz antenna design. This architecture of end-fire planar rod antenna has very simple architecture and meets the requirement of the challenges of on-chip antenna design.
Dear reviewer, in line 282, I have added a sentence to illustrate the problem:
Especially this type of planar end-fire antenna architecture based on SIW meets the requirement that on-chip antennas demand simplicity in structure and it is possibly suitable for applications like integrated planar THz antenna design.
- The authors should clarify additional performance metrics, such as radiation efficiency and sidelobe levels, in the results section.
Response: Thank you sincerely for your suggestions. In line 238 I have added the results of radiation efficiency and sidelobe levels, as follows:
Radiation efficiency is 0.58, and sidelobe level in x-y plane is -5 dB.
- More recent references should be added.
Response: Thank you sincerely for your suggestions. Dear reviewer, because most of the cylindrical rod antennas are not so up to date, now I add 5 more recent references about cylindrical rod antennas in line 65-74: references [19-23], marked with red color. Sincerely hope that now it could be better.
- Figures are not clear enough.
Response: Thank you sincerely for your suggestions. Now I have already made all the figures bigger , as big as possible, and sincerely hope that now it is better.
- The writing style is not good.
Response: Thank you sincerely for your suggestions. Now I have modified my paper here and there which is marked with red color, sincerely hope that now it could be better in writing style .
- What is the meaning of Figure 8? Please clarify it.
Response: Thank you sincerely for your suggestions.
Figure 6 gives the top view, bottom view and side view of the prototype of the end-fire rod antenna in 1st fabrication and measurement. As shown in figure 6(a), GSG coplanar waveguide with four small rectangular vias is designed as input for the in-stallation of the end launch connector during measurement. To measure the end-fire rod antenna, 2.92mm end launch connector fabricated by Qualwave Inc, China is used, as shown in figure 6(a).
Figure 8 gives the top view, bottom view and side view of the prototype of the end-fire rod antenna in 2nd fabrication and measurement. As shown in figure 8(a), GSG coplanar waveguide without the 4 small rectangular vias is designed as input for the installation of the 2.92 mm end launch connector during measurement.
Comparing figure 6 and figure 8, we can see that in 2nd fabrication and measurement, GSG coplanar waveguide without the 4 small rectangular vias is designed as input for the installation of the 2.92 mm end launch connector during measurement.
(a)
(b) (c)
|
Figure 6. (a)top view (b) bottom view and (c)side view of prototype of the end-fire rod antenna in 1st fabrication (coordinate X-Y-Z axis is also shown in (c)).
(a) (b) (c) |
Figure 8. (a)top view (b) bottom view and (c)side view of prototype of the end-fire rod antenna in 2nd fabrication (coordinate X-Y-Z axis is also shown in (c)).

Round 2
Reviewer 1 Report
Comments and Suggestions for Authors
The authors have addressed the comments.
Reviewer 3 Report
Comments and Suggestions for Authors
The authors have addressed my concerns.